# Self-Supervised Representation Learning for High-Content Screening

**Daniel Siegismund**[*]                                   DANIEL.SIEGISMUND@GENEDATA.COM
**Mario Wieser**[*]                                           MARIO.WIESER@GENEDATA.COM
**Stephan Heyse**                                            STEPHAN.HEYSE@GENEDATA.COM
**Stephan Steigele**                                       STEPHAN.STEIGELE@GENEDATA.COM
*Genedata AG, Basel, Schwitzerland*

**Editors:** Under Review for MIDL 2022

## Abstract

Biopharma drug discovery requires a set of approaches to find, produce, and test the safety of drugs for clinical application. A crucial part involves image-based screening of cell culture models where single cells are stained with appropriate markers to visually distinguish between disease and healthy states. In practice, such image-based screening experiments are frequently performed using highly scalable and automated multichannel microscopy instruments. This automation enables parallel screening against large panels of marketed drugs with known function. However, the large data volume produced by such instruments hinders a systematic inspection by human experts, which consequently leads to an extensive and biased data curation process for supervised phenotypic endpoint classification. To overcome this limitation, we propose a novel approach for learning an embedding of phenotypic endpoints, without any supervision. We employ the concept of archetypal analysis, in which pseudo-labels are extracted based on biologically reasonable endpoints. Subsequently, we use a self-supervised triplet network to learn a phenotypic embedding which is used for visual inspection and top-down assay quality control. Extensive experiments on two industry-relevant assays demonstrate that our method outperforms state-of-the-art unsupervised and supervised approaches.

## 1. Introduction

In recent years, there has been tremendous progress in the development of novel drug and treatment strategies in various disease areas such a vaccines (Polack et al., 2020) or cancer immunotherapies (Kruger et al., 2019).

One driver here, High-Content Screening, based on automated microscopy and lab automation has become an important toolbox to mitigate this trend by systematically analyzing the drug candidates in thousands of cells to develop new treatments. A good high content assay allows to systematically measure complex phenotypes triggered by application of drug candidates. Therefore, a primary goal is achieving a highly reliable assay quality which reflects the biological change of the cells and ensures the selection of relevant drug candidates for further analysis.

To date, the analysis of High-Content Screening assays is performed by handcrafted feature-based analysis (Caicedo et al., 2017) using classical image analysis software e.g. Cellprofiler (Carpenter et al., 2006) or deep learning based approaches (Godinez et al., 2017; Steigele et al., 2020; Dürr et al., 2018). However, all of the aforementioned approaches require labels or other prior knowledge to conduct the often difficult analysis, biased by the curating scientist designing the analysis workflows.

---

[*] Contributed equally

To overcome these limitations, we propose a novel data curation approach for High-Content Screening without any supervision, where we employ the concept of archetypal analysis (Cutler and Breiman, 1994) to extract phenotypes in an unsupervised fashion. In contrast to high-dimensional clustering techniques or principal component analysis, archetypal analysis is grounded on biological principles of phenotype discovery (Shoval et al., 2012; Tendler A, 2015). Here the idea is, that each individual is a convex mixture of pure phenotypes or archetypes - allowing to uncover the important phenotypes, by employing a triplet network to learn an embedding that group similar phenotypes/archetypes together. This embedding may subsequently be used for visual inspecting and as stand-alone, complete top-down analysis including assay quality control.

Our contributions are summarized as follows:

- Extraction of biologically relevant phenotypes in an unsupervised fashion based on archetypal analysis.

- Learning of an embedding given the extracted phenotypes by employing self-supervised learning.

- Using industry relevant screening datasets demonstrate that the proposed model even outperforms state-of-the-art supervised approaches.

## 2. Related Work

**Archetypal Analysis**  Archetypal Analysis aims to identify extremal points in a given dataset (Cutler and Breiman, 1994). As the traditional approach deals merely with static data, Stone and Cutler (1996) and Cutler and Stone (1997) proposed an extension to deal with time-resolved data. More recently, Prabhakaran et al. (2012) discovered the open problem of manual model selection and introduced an approach to automatically select the optimal number of archetypes via group-lasso constraints. In addition, the classical approach suffers from limited model flexibility as it assumes linearity, this is why Bauckhage and Manshaei (2014); Mørup and Hansen (2012) introduced a kernel-based archetype approach whereas Keller et al. (2021, 2019) proposed to model non-linearity with neural networks based on the information bottleneck principle (Tishby et al., 1999; Alemi et al., 2017; Wieczorek et al., 2018). Moreover, archetypal analysis assumes that the data is real valued which is not reasonable in many situations. As a consequence, Seth and Eugster (2016) provided a probabilistic formulation of the archetype problem that is able to deal with different data types. However, this formulation makes strong assumptions on the underlying data distribution. Hence, Kaufmann et al. (2015) proposed a copula formulation that relaxes these assumptions by only assuming a Gaussian dependency structure with arbitrary margins that is also able to deal with mixed and missing data.

**Metric Learning**  Quantifying the distance between data samples or feature vectors is a challenging task for many machine learning algorithms. In certain cases, we employ prior domain knowledge to select a predefined metric that is subsequently used by our algorithm. However, in many cases it is unreasonable to employ a standard metric for certain tasks and datasets. As a consequence, metric learning attempts to overcome these problems by learning a suitable distance metric for such specific tasks. More specifically, Hoffer and Ailon (2015) introduced the concept of triplet networks. This approach selects a positive and a negative example per data point that aims to push

the negative example far away while simultaneously contracting the positive example to the current data point. Inspired by the success of the aforementioned approach, Schroff et al. (2015a) applied these ideas to improve face recognition systems in the context of computer vision. Despite its success, current approaches of metric learning have the limitation that they treat negative examples equally which may lead to suboptimal results. For this reason, Zhou et al. (2020) introduced a ladder loss to account for the relevance of the negative examples. In its current form, such approaches have been shown to be highly powerful for learning class-discriminative distance metrics but to have limited abilities for learning diverse data characteristics. Milbich et al. (2020) circumvent these limitations by introducing a feature aggregation loss which leads to improved generalization abilities in metric learning, overall.

Consequently, researchers in High-Content Screening (HCS) have been inspired by the success of the aforementioned metric learning approaches and applied these concepts to various HCS tasks. Here, we can distinguish between supervised and unsupervised approaches for HCS tasks: In the supervised case, metric learning has been used to classify the mechanism of action (MOA) by using a triplet loss (Ando et al., 2017) instead of applying a supervised classification model. This work is further been extended by Caicedo et al. (2018) which aimed to classify MOAs using single-cell feature embeddings that solely rely on weak instead of strong supervision. Moreover, these metric learning approaches have been used to perform automating morphological profiling based on Generic Deep Convolutional Networks (Pawlowski et al., 2016). In the unsupervised case, Janssens et al. (2020) transferred to knowledge of unsupervised metric learning to the context of MOA classification in high-content cellular images by employing a fully unsupervised deep mode of action learning approach. Concurrent work has been proposed by Lafarge et al. (2019) who also classified MOAs using an unsupervised metric learning method. Last, Perakis et al. (2021) improved treatment classifications upon supervised approaches by using self-supervised contrastive learning (He et al., 2020; Chen et al., 2020b) to obtain better representation of single-cell phenotypes.

In contrast to the aforementioned approaches, our work focuses on learning meaningful representations for downstream analysis based on archetypal self-supervision.

## 3. Model

We introduce a three step approach to extract relevant phenotypes in High-Content Screening. In the first step, we extract pseudo-labels based on archetypal analysis. Subsequently, we learn an embedding in self-supervised fashion based on the previously extracted pseudo-labels. Afterwards we perform a top-down analysis of the learned embedding to conduct downstream analysis tasks e.g. assay quality statistics. For this analysis, we take the features from the embedding layer as input features.

**Learning of Low-Dimensional Feature Map of Cell Images**    Given input image $I$, we extract a feature map $F$ by using a pretrained convolutional neural network $c$. Specifically, we take the last flattened layer as feature map $F \in \mathbb{R}^{1 \times p}$ where $p$ denotes the number of feature dimensions.

**Extract Label Information via Archetypes**    In this step, we take the previously calculated feature maps $F$ and perform the archetype-based pseudo-label assignment. As previously described in Section A.1, archetypal analysis is a method to extract extreme data points from a dataset. To do so, we build a data matrix $X \in \mathbb{R}^{n \times p}$. This matrix consists of $n$ feature maps with $p$ dimensions.

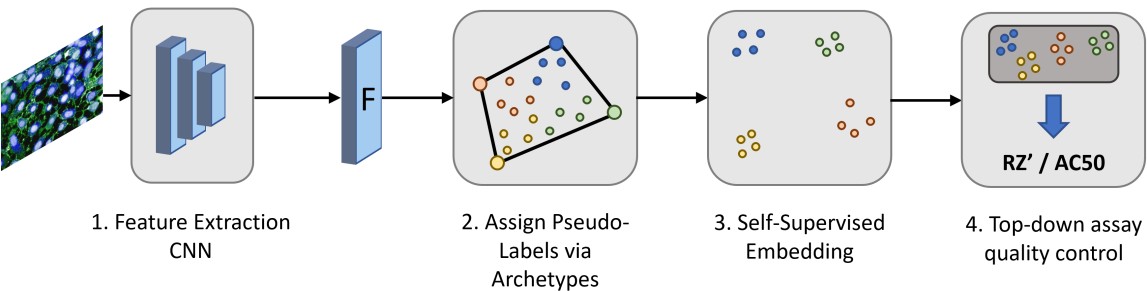

Figure 1: **Self-supervised representation learning approach for High-Content Screening** Black arrows denote steps and gray boxes actions in our workflow, respectively. In the first step (1.), we extract a low-dimensional feature representation F from an image by employing a CNN backbone. In the subsequent step (2.), we learn an archetypal representation of the feature representation F and pseudo-label each data point by assigning it to its closest archetype. In the third step (3.), we learn a self-supervised embedding based and use this embedding to perform top-down assay analysis tasks such as quality control (4.).

Our goal to decompose the feature matrix $X$ into the weight matrices $A \in \mathbb{R}^{n \times k}$ and $B \in \mathbb{R}^{k \times n}$ as well as into an archetype matrix $Z \in \mathbb{R}^{k \times p}$ where $k$ denotes the number of archetypes. Here, it is particularly important that the rows of $A$ and $B$ are row-stochastic. More intuitively, row-stochasticity means that the entries of a row sum to one:

$$a_{ij} \geq 0 \ \wedge \ \sum_{j=1}^{k} a_{ij} = 1 \qquad b_{ji} \geq 0 \ \wedge \ \sum_{i=1}^{n} b_{ji} = 1 \tag{1}$$

As a result, we are able to represent our data matrix $X$ as a weighted sum of archetypes $Z$ as well as the archetypes as a weighted sum of the data matrix $X$.

$$\mathbf{x}_i \approx \sum_{j=1}^{k} a_{ij} \mathbf{z}_j = \mathbf{a}_i Z \qquad \mathbf{z}_j = \sum_{i=1}^{n} b_{ji} \mathbf{x}_i = \mathbf{b}_j X \tag{2}$$

To this end, our problem boils down to learn the correct weight matrices $A$ and $B$ given our feature matrix $X$ and the predefined number of archetypes $k$. The central problem of AA is finding the weight matrices $A$ and $B$ for a given data matrix $X$ and a given number $k$ of archetypes. Hence, the optimization problem consists in minimizing the following objective function:

$$\min_{\mathbf{A}, \mathbf{B}} \| \mathbf{X} - \mathbf{A} \mathbf{B} \mathbf{X} \|^2 \tag{3}$$

After having learned the weight matrices $A$ and $B$, we take all images $I$ to pseudo-label our data set for representation learning. Specifically, we take weight matrix $A$ to select the most prominent archetype as label by employing a softmax function:

$$l_i = argmax(\frac{\exp(a_i)}{\sum_j \exp(a_j)}) \tag{4}$$

This results in a fully labeled data set for each cell image which we subsequently use to create triplets for the self-supervised representation learning part.

**Self-Supervised Representation Learning** In this work, we adopt a self-supervised representation learning approach based on triplet losses to obtain an embedding $E$ (Schroff et al., 2015a). Here, the main idea is to group similar cells together while pushing dissimilar cells away. Our embedding $E$ is represented by a convolutional neural network $g(I)$ which takes the image $I$ as an input and maps this image into a d-dimensional latent representation.

As this approach is fully self-supervised, we have to provide the algorithm with image triplets to learn the representation $E$. The triplets consist of there different images: an anchor image $f^a$, a similar (positive) image $f^p$ and a completely dissimilar (negative) image $f^n$. As a result, we optimize the following loss function to obtain the embedding $E$:

$$\sum_{i=1}^{N}\Big[||f_i^a - f_i^p||_2^2 - ||f_i^a - f_i^n||_2^2 + \alpha\Big] \tag{5}$$

where $\alpha$ denotes the margin between positive and negative pairs. As we are in an unsupervised setting, we determine positive and negative images for the triplets by making use of pseudo-labels obtained by the archetypal analysis from the previous section.

**Downstream Analysis** To assess the quality of the computed self-supervised embedding, the features from the embedding layer for every cell image are extracted. For qualitative evaluation the extracted features are visualized via t-SNE (Van der Maaten and Hinton, 2008). For quantitative assessment we apply standard procedures for downstream analysis in High-Content Screening using additional experimental annotations (e.g. control conditions) (Bray and Carpenter, 2017). This is done for each feature separately. Further details of the computed downstream metrics are discussed in Section 4.

## 4. Experiments

Real-world HCS datasets are tested by their quantitative and qualitative performance: The NTR1 (Peddibhotla et al., 2013) dataset is used for a qualitative assessment of the self-supervised embedding. The BBBC013 (Ljosa et al., 2012) dataset is used to evaluate the quantitative performance of the embedding features with regard to assay quality metrics.

### 4.1. NTR1 Dataset

**Dataset and Setup.** The data set consists of 36864 images which are obtained from an internalization assay of the neurotensin receptor 1 (NTR1). More specifically, this data stems from a screen for modulators of NTR1. Upon activation this G-protein-coupled receptor is internalized into endosomes in a beta-arresting mediated process. Here, the redistribution of $\beta$-arrestin-conjugated green fluorescent protein (GFP) was measured to assess the activation of NTR1. Further details can be found in the original publication (Peddibhotla et al., 2013).

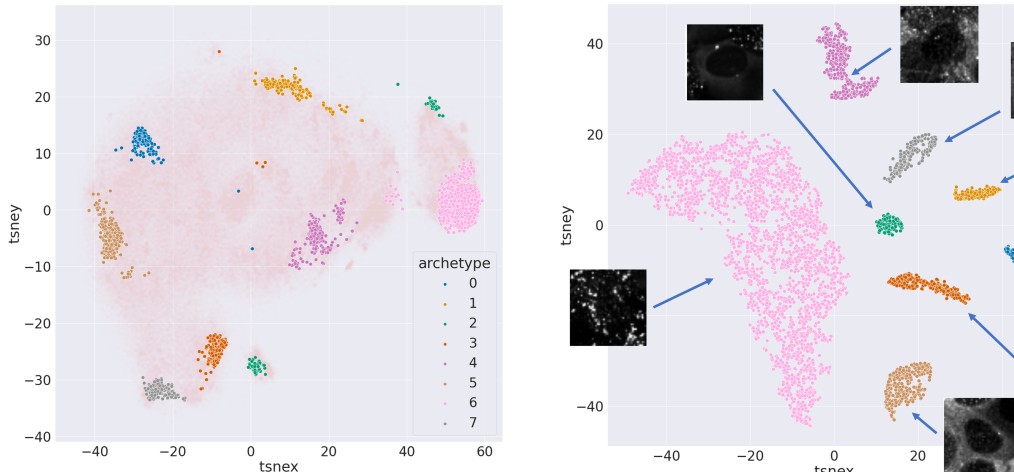

Figure 2: **Overview of two different embeddings produced for Dataset 1 (NTR1):** Left: embedding of the complete data using the extracted features - highlighted are the most archetypical examples of the calculated eight archetype classes., Right: Self supervised embedding of the eight archetype classes.

**Metrics.** The learned embedding is qualitatively tested with respect to the estimated archetypes.

### 4.2. BBBC013 Dataset

**Dataset and Setup.** This dataset encompasses 192 images from Human U2OS cells showing cytoplasm to nucleus translocation of the Forkhead (FKHR-EGFP) fusion protein. Blocking PI3 kinase / PKB signaling with Wortmannin (Cpd 1) or with the compound LY294002 (Cpd 2) leads to accumulation of FKHR-EGFP within the nucleus. Both compounds are applied in nine point dose responses in addition to negative and positive controls. Nuclei are stained with DRAQ. Further details about the assay may be found with accession number BBBC013 in the Broad Bioimage Benchmark Collection (Ljosa et al., 2012).

**Baselines.** Our method is benchmarked against two unsupervised clustering methods that provide pseudo-labels for the training of the self-supervised embedding using triplet networks: k-Means clustering and Gaussian Mixture Models (details in Appendix C). In addition, we benchmark our method against a recent state-of the art self-supervised approach with successful application for biomedical images - SimCLR (Chen et al., 2020a; Chaitanya et al., 2020) and a complete supervised retraining of the EfficientNetB2 using ground-truth labels. The performance of our method is compared to previous published work on the same dataset using either conventional image analysis (Carpenter et al., 2006) or a supervised training with a multi-scale deep neural network (Godinez et al., 2017); however, both studies do not report the complete metrics. In addition, we report the performance of the embedding from the EfficientNetB2 without further self-supervision.

**Metrics.** We assess the maximum Z' factor (max Z') as performance measure (Zhang et al., 1999). The Z' factor measures the statistical effect size to judge the quality of a biological assay - a standard

in high throughput screening. The values can range between 1 and -inf with a good assay quality if the Z' factor is larger than 0.5. For the calculation the data of negative and positive controls are used. In this study the Z' factor for all features from the self-supervised embedding is used and only the maximum value is reported. The second metric applied for this dataset is the half maximal effective concentration ( $EC_{50}$ ) which describes the necessary compound concentration to obtain the half response between two control stages. The $EC_{50}$ value (also called potency) is one of the key result parameters in drug discovery (Sebaugh, 2011). Z' factor and $EC_{50}$ are calculated using Genedata Screener. For a more detailed decription of the applied metrics please see Appendix D.

| Method | max Z' | Cpd1 $EC_{50}$ | Cpd2 $EC_{50}$ |
|---|---|---|---|
| Archetypes w/o self-supervised embedding (pseudo-labeled extraction network features) | 0.540 | 12.0 | 4.3 |
| EfficientNetB2 supervised | 0.935 | 8.4 | 3.6 |
| Carpenter et al. (Carpenter et al., 2006) | 0.910 | 9.0 | − |
| Godinez et al. (Godinez et al., 2017) | − | 7.7 | 2.3 |
| GMM* | −2.204 | 91.5 | 35.2 |
| k-Means* | −1.888 | no fit | no fit |
| SimCLR* | −5.424 | no fit | no fit |
| **Archetypes (Ours)*** | **0.943** | 11.9 | 5.5 |

Table 1: Quantitative summary of results from BBBC013 experiments. Triplet Networks with GMM, k-Means and archetypal pseudo-labeling, SimCLR, and supervised approaches. For all models the max Z' and $EC_{50}$ values are considered. max Z' close to one and $EC_{50}$ close to supervised solution is better. Archetypes outperform even supervised approaches on max Z' and are comparable in terms of $EC_{50}$. The * denotes self-supervised methods.

## 5. Discussion

The idea of archetypal analysis is based on finding extremal points in multidimensional data sets instead of uncovering cluster centers. This could render it into a potent approach for real-world applications where the analysis of such extremals is one goal. This makes this approach appropriate for phenotypic drug-discovery as High-Content Screening focuses on extremal phenotypes frequently, e.g. control phenotypes (Moffat et al., 2017).

**Archetypical Self-Supervised Representation Learning Produces Meaningful Embeddings.** In Figure 2 the embedding using archetypical self-supervision is shown for the NTR1 dataset. The left side of Figure 2 shows the t-SNE embedding of the feature outputs from the EfficientNet (see Appendix B). The extracted eight archetypes are highlighted in different colors whereas the other cells are shown translucent. The algorithm is able to identify different archetypes that would be not just found using pure t-SNE embeddings. One such example is the green archetype which is located in different locations in the map. On the right side of Figure 2 a t-SNE embedding of the learned self-supervised features is displayed in conjunction with an exemplary cell image per archetype class. Utilizing this map, control phenotypes (pink and grey archetypes) and novel phenotypes (e.g. red archetype corresponding to overexpression) can easily be identified. For further information on

the prevalent phenotypes that occur in the NTR1 assay please see (Siegismund et al., 2021).

**Archetypical Self-Supervised Representation Learning Outperforms in Downstream Analysis Tasks.** In addition to the qualitatively improved embeddings, our proposed method outperforms different state-of-the-art self-supervised representation learning techniques, as well as complete supervised techniques on High-Content downstream analysis tasks. Indicative is the max Z', a standard quality indicator for characterizing a batch of biological measurements (e.g. a single experimental unit = one microtiter plate) compared to results from published supervised and unsupervised approaches (Table 4.2 shows the max Z' and the $EC_{50}$ values (please see 4.2)). The proposed approach receives the highest max Z' of 0.943, close to the optimal value of 1. This is in contrast to the self-supervised competitors: GMM, k-Means and SimCLR which achieve quite worse max Z' values of -2.204, -1.888 and -5.424 respectively; well below any acceptable quality threshold of 0.5 (Zhang et al., 1999). Interestingly, the new approach also outperforms supervised approaches that rely on labeled data (0.935); and methods using prior knowledge (0.910 - (Carpenter et al., 2006)). In addition, we performed an ablation experiment where we directly calculated max Z' on the archetypal pseudo-labeled features from the extraction network (EfficientNet unsupervised) without learning the self-supervised embedding. Here, we achieve a max Z' value of 0.540. Thus, a subsequent self-supervised embedding step using archetypes dramatically improves the result.
The $EC_{50}$ values for Cpd1 and Cpd2 show close values to the ones calculated from the supervised methods. Of special note is that for all other self-supervised methods no meaningful results were produced, and for k-means and SimCLR no concentration response fitting could be obtained (more details in Appendix D). Overall, we found that our proposed approach can produce biologically more relevant features for downstream analysis than other self-supervised approaches.

## 6. Conclusion

High-Content Screening, using automated microscopy forms a crucial drug discovery technology for the systematic analysis of thousands of drug candidates and for a deep characterization of lead candidates in developing drugs. So far, workflows rely on labels and on prior knowledge to drive the data analytics and obtained insights. This process, however, does not scale enough and often results in highly expensive data curation processes. In this work, we demonstrate that archetypes can be defining elements to uncover important pharmacological phenotypic endpoints; without relying on any prior knowledge. The method is fully self-supervised and hence can be combined with other workflow elements, e.g., by presenting a panel of learned archetypes to a human curator via active learning. As a consequence, learned representations based on archetypal analysis have a large potential for further automation of existing real-world deep-learning analysis workflows as they often rely on a expensive, manual and biased training data curation process (please see (Steigele et al., 2020) for a exemplary workflow).

In summary, we demonstrated in this work that we can extract relevant and meaningful biological phenotypes using archetypal analysis, learn embeddings for visual inspection and for performing a practical top-down analysis incl. assay quality control; outperforming state-of-the-art approaches on two industry relevant screening datasets, considering practical standard metrics from drug screening.

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

## Appendix A. Foundations of Archetypal Analysis

### A.1. Formal Definition

Archetypal analysis (AA) was first proposed by Cutler and Breiman (1994). It is a linear procedure where archetypes are selected by minimizing the squared error in representing each individual data point as a mixture of archetypes. Identifying the archetypes involves the minimization of a non-linear least squares loss. More specifically, linear AA is a form of non-negative matrix factorization where a matrix $X \in \mathbb{R}^{n \times p}$ of $n$ data vectors is approximated as $X \approx ABX = AZ$ where $A \in \mathbb{R}^{n \times k}$ and $B \in \mathbb{R}^{k \times n}$ represent the weight matrices with $k$ representing the number of archetypes, $n$ the number of data samples and $p$ the dimensionallity of the feature matrix $X$, respectively. In addition, we denote $Z \in \mathbb{R}^{k \times p}$ the archetype matrix. In general, we assume that $k < \min\{n, p\}$. In order to solve this non-linear optimization problem, we aim to minimize the following residual sum of squares:

$$\min_{\mathbf{A}, \mathbf{B}} \|\mathbf{X} - \mathbf{ABX}\|^2 \tag{6}$$

### A.2. Intuition and Comparison to Clustering Methods

We provide a more intuitive explanation for Archetypal Analysis to demonstrate the functionality and the differences compared to classical clustering approaches (see Figure A.1). To do so, we simulate a two-dimensional dataset where we define three archetypes at positions (0,0), (1,0) and (0.5,1.73). Subsequently, we sample 250 combinations by drawing samples from a Dirichlet distribution with parameters (1,1,1) which corresponds to matrix A from the AA method. To obtain our data points, we multiply A with our archetypes and add Gaussian noise with variance 0.005.

In clustering approaches such as k-means the goal is to find prototype points in the dataset, called cluster centers. However, in certain cases no cluster structure is present like in the case of many phenotypic driven drug screening problems. For example, in Figure A.1(*a*) k-means is not capable in finding reasonable prototypes that describe the structure of the data distribution. In such cases, it is intuitive to look at archetypes because this method does not look for prototypes but for extremal data points that form a convex hull. More specifically, all data points are described as a convex combination of the archetypes which you can see in Figure A.1(*b*). The more archetypes we add for our analysis the better is the approximation of the convex hull.

## Appendix B. Algorithmic Implementation

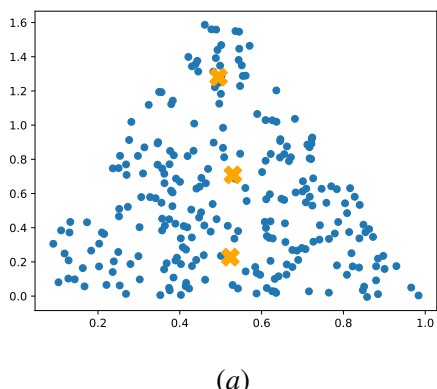 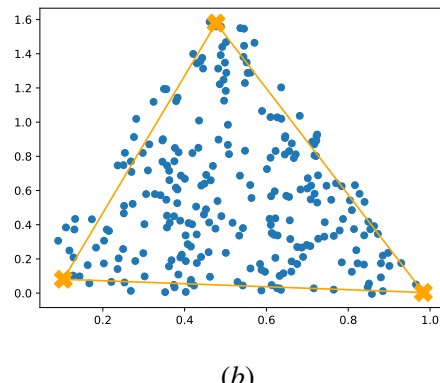

(a)                                          (b)

Figure A.1: A comparision of k-means clustering and archetypal analysis. Blue dots denote data samples. In Figure A.1(*a*) the orange cross denote the cluster means found by k-means clustering. In Figure A.1(*b*) the cross illustrates found archetypes and the orange line the convex hull.

---

**Algorithm 1** Archetypal Self-Supervised Representation Learning
___
**Input:** Set of images $I$
**Output:** embedding $e$
  1: extract features F from image I with CNN c(I)
  2:
  3: take F and calculate archetypes A
  4: assign every feature map F to closest archetype A and use as pseudo-labels
  5:
  6:
  7: **for** each epoch **do**
  8:     sample $i$ minibatches of $x$, $y$ and $t$
  9:     **for** each minibatch $i$ **do**
 10:         encode $x_i$ into $p_\eta(z_i \mid x_i)$
 11:     **end for**
 12: **end for**
___

| Dataset | $k$ (# of archetypes) | Dense layers | learning rate | dropout rates | batch size | epochs |
|---------|------------------------|--------------|---------------|---------------|------------|--------|
| BBBC013 | 4 | [500 50 24] | 0.001 | [0.3 0.3] | 64 | 400 |
| NTR1 | 8 | [500 50 6] | 0.001 | [0.15 0.22] | 128 | 400 |

Table B.1: Overview of the used hyperparameters used for the training of the triplet network for both datasets

## Appendix C. Experimental Setup of Baseline Methods

In the following section, we describe the experimental setup for the benchmark methods used in this study.

### C.1. Feature Extraction Process

The extraction of single cell images from the original images is performed on Genedata Imagence (Steigele et al., 2020). These single cell images have a size of 50x50 px (Dataset 1) or 120x120 px (Dataset 2). As feature extraction network we used an EfficientNetB2 (Tan and Le, 2019) pretrained on Imagenet via Noisy-Student (Xie et al., 2020). The penultimate layer of the network was used as output (1408 features).

### C.2. SimCLR

For the encoder architecture, we defined a convolutional neural network with four convolutional layers with kernel size = 3 and strides = 2, followed by a fully-connected layer with 128 neurons and relu activation functions. Furthermore, we defined a non-linear projection head with 128 neuron and a relu activation function. We trained our model for 400 epochs with a batch size of 525 by employing an Adam optimizer with learning rate=0.001, beta1=0.9 and beta2=0.999. For augmentation, we performed standard augmentation, namely flipping, translation, zooming and jitter.

### C.3. GMM and k-means

As parameters for the clustering algorithms the default values for their respective implementation from the scikit-learn framework are used (Pedregosa et al., 2011). The cluster number is set to the number of archetypes determined via optimization to ensure comparability. Afterwards the exact procedure as for the archetype applying the same hyperparameters for the datasets (Table B.1) is performed to obtain the self-supervised embedding.

### C.4. Archetypal Analysis

Analysis of the archetypes is done via coreset approximation on the embedding vectors for all cell images (Mair and Brefeld, 2019). Afterwards, pseudo-labels are extracted from the archetypical composition vector of each cell image.

### C.5. Self-Supervised Triplet Network

The learning of the self-supervised embedding is done via a retraining of the EfficientNetB2 adding three Dense and two Dropout layers using Triplet Semi Hard loss (Schroff et al., 2015b). The determined pseudo labels are used as class labels for the network retraining. For parameter and hyperparameter tuning (dropout rates, size of dense layers, learning rate and the optimal number of archetypes $k$) the optuna hyperparameter optimization framework was applied with the triplet loss as objective (Akiba et al., 2019). Table B.1 shows the used hyperparameters as determined with optuna optimization for both datasets.

## Appendix D. Details on the evaluation metrics

In contrast to the performance metrices common for the field of machine learning e.g., classification accuracy we use two domain specific and very practical metrics ($EC_{50}$ and max Z'), predominant for biopharma industry.

First, the Z' factor is a metric to investigate the quality of screening assays e.g., if the controls are suitable to answer the particular biological questions. It is defined as:

$$Z' = 1 - \frac{3 \cdot (\sigma_p + \sigma_n)}{|\mu_p + \mu_n|} \tag{7}$$

Where $\sigma_p$ and $\mu_p$ denote the sample standard deviation and sample mean of the positive (p) controls. Whereas the subscript n refers to the neutral controls.

The here interesting point is, that there is no linear correlation between classification accuracy and Z' factor (Kümmel et al., 2010). Thus, classification accuracies cannot be used as a suitable proxy to describe assay quality.

Second, the 50 % effective concentration $EC_{50}$ describes the needed dosage of a drug where the biological effect is 50 %. It's really important ot note, that the $EC_{50}$ is likley the most important and thus most heavily used metric in any screening assay to assess aiming to assess drug potency. The calculation of the $EC_{50}$ is commonly done via fitting the Hill equation to the measured data points with:

$$Y = S_0 + \frac{(S_{inf} + S_0)}{1 + (\frac{EC_{50}}{[C]})^n} \tag{8}$$

$S_0$ is the fitted activity level at zero concentration of test compound ("zero activity"); $S_{inf}$ is the fitted activity level at infinite concentration of test compound ("infinite activity"); $n$ is the Hill coefficient for the curve, i.e. the measure of the slope at $EC_{50}$; $[C]$ is the concentration and $Y$ the activity.

Figure D.1 shows an exemplary dose-response curve with all parameters described in Equation 8. Please note that the data cannot always be fitted with Equation 8 e.g. if $S_0$ and $S_{inf}$ possess the same value (no activity). Thus no $EC_{50}$ value can be determined and has been denoted as "no fit" in Table 4.2. The value of the $EC_{50}$ should be ideally close to the theoretically value of the compound (difference less than one order of magnitude).

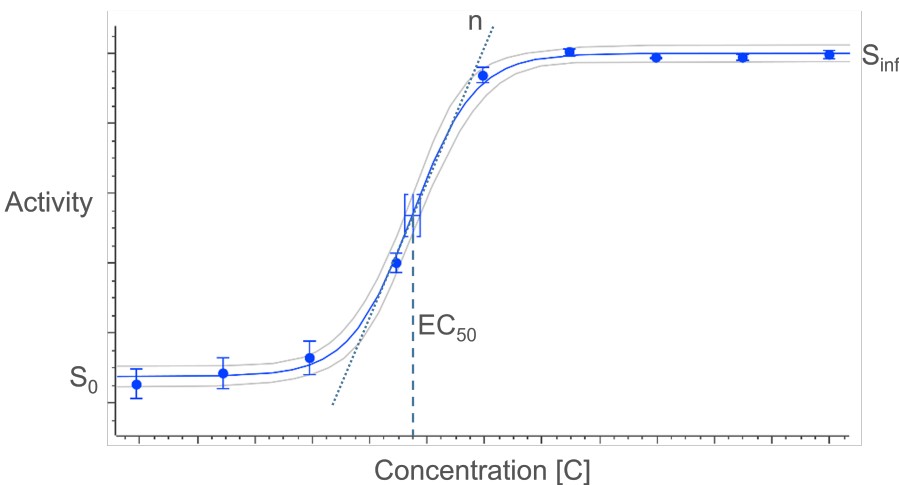

Figure D.1: Exemplary hill fit of a drug dose–response relationship with all important parameters. $S_0$ and $S_{max}$ denote the activity level at zero and infinite concentration respectively; the hill parameter $n$ defines the slope at $EC_{50}$ and $EC_{50}$ the effective concentration at the 50% activity level.

