# OpenReview forum: "Self-Supervised Representation Learning for High-Content Screening"
_MIDL.io/2022/Conference — MIDL 2022_

### Official Review · Reviewer_HiST · 2022-01-20

**Confidence:** 2
**Preliminary Rating:** 2
**Recommendation:** Poster

**Summary:**

The paper proposes an unsupervised approach for learning phenotype embedding. The learned embedding is used to generate pseudo-labels, which are subsequently used for self-supervision. The embedding learned using self-supervision shows better performance on two real-world industrial assays datasets for downstream visual inspection and quality control. Overall, the paper is novel with good experimental set-up and results.

**Strengths:**

* The paper tackles an important real-world problem associated with drug discovery. As per my knowledge, not many papers in the field of medical image analysis have tackled this problem.
* The paper works on real-world industrial datasets and shows the usefulness of the proposed method.
* The paper gives a really good literature review on all relevant topics (Ex. Archetypal Analysis, Metric learning including self-supervision).
* The results show that the proposed method gives better performance compared to even supervised methods on relevant evaluation metrics.


**Weaknesses:**

* It was not clear how archetypal analysis gives extreme data points from a dataset. As AA is not a commonly employed method in medical image analysis, a more detailed description with a small visualization on a toy dataset in the appendix would help to improve understanding.
* It was not clear why softmax was applied on the weight matrix B. Matrix B is of size $ k * n $, where $k$ is the number of archetypes and $n$ is the number of images. I am assuming that softmax is applied on the first dimension of the matrix ($k$), which will allow the authors to assign a single archetype to each image $n$ using the $argmax$ operator.  On the other hand, the weight Matrix A is of dimension $n * k$. We also have approximation $X \approx A*Z$ and $\sum_{j=1}^k a_{ij} = 1$. So simply applying $argmax$ row-wise could give us a single archetype of each image. A good explanation for this would be helpful.
* In the whole paper or in the appendix, it is not mentioned what was the chosen value of $k$. Was this consistent across both datasets? Did the performance vary too much with a change in $k$? Was the same $k$ value used for all unsupervised clustering methods (PCA and GMM)?
* It is not clear, why a self-supervision network is necessary when Archetypal analysis already provides a good clustering of features.
* In the second experiment section, the proposed method is compared against GMM, K-means, and SimCLR methods. Why the analysis using only the archetypal analysis is not provided? I am assuming the reported archetypes (ours) is a combination of archetypal analysis followed by self-supervision.
* How to interpret Cpd1 EC 50 and Cpd2 EC 50 values? Is higher value better compared to a lower value?
* It was not discussed why for k-means and SimCLR no concentration-response fitting was obtained which results in "no fit" for Cpd1 EC 50 and Cpd2 EC 50.

**Deanonymize Review:**

no

**Detailed Comments:**

* Maybe use different color coding for clustering as it was not easy to identify red-colored cluster mentioned in the discussion (novel phenotype)
* Size of features used for archetypal analysis and self-supervision is not mentioned clearly in the paper or the appendix. The only mention is that the penultimate layer of efficientnetB is used for archetypal analysis and additional dense layers are added for self-supervision.
* For compared supervised approaches, how the labels were generated for training purposes?
* There are some typos throughout the paper. Ex. Page-7, "The values can range between 1 and -inf with a good assay quality
if the Z’ factor is larger 0.5" is missing "than".


**Final Rating After The Rebuttal:**

4: Weak Accept

**Justification Of The Final Rating:**

The authors have provided sufficient clarification and new experiments to alleviate most of my concerns; There are still a couple of points that need to be addressed. I have provided them in my official comment below. These points can be quickly addressed during the final version of the paper. Keeping this in mind, I would like to change my initial rating and recommend a weak acceptance of the paper. The main reason for not providing strong acceptance is that the paper is not exactly in the line of MIDL.

**Paper Type:**

both

**Questions To Address In The Rebuttal:**

* Mainly points 2,4, and 5 from the weakness section need to be addressed, as they put the methodology of the paper in question.
* Addressing other mentioned points in both weakness and detailed comment section can improve the clarity of the paper.

**Special Issue:**

no

---

### Official Review · Reviewer_YCk1 · 2022-01-24

**Confidence:** 3
**Preliminary Rating:** 4
**Recommendation:** Oral, Poster

**Summary:**

The authors address the task of Self-Supervised Representation Learning for High-Content Screening. Their pipeline comprises of three steps - (1) generating unsupervised pseudo labels for image data using *Archetype Analysis*, (2) training a network for triplet loss using the acquired labels from the previous step and generating representations for images, and (3) performing downstream analysis and visualization.  The use of Archetype Analysis to generate labels is a novel contribution!

**Strengths:**

- In general, the introductory text, the related work section about metric learning on HCS  and *Figure One* indicating the pipeline were well done and easy to follow.

- The use of Archetype Analysis to obtain pseudolabels followed  by representation learning appears as a novel contribution to me (I am not aware of other works using AA in this fashion).


**Weaknesses:**

- There is a related unsupervised approach that operates by iteratively generating pseudolabels and then producing embeddings called [SwAW](https://arxiv.org/pdf/2006.09882.pdf) which could be considered as a direct baseline method on account of it producing pseudolabels as well. Comparison with respect to *SwAW* would be welcome.

- The Section 3 detailing Archetype Analysis was not completely clear to me. This section could be expanded and more details provided.



**Deanonymize Review:**

yes

**Detailed Comments:**

- >"Our contributions summarizes as follows:"

I would suggest rewording this to be "Our contributions are summarized as follows:" or "We summarize our contributions as follows"

- > "Archetypal Analysis aims to identify extermal points in a given dataset"

Small typo: should be *extremal* instead of *extermal*

- > " In the supervised case, metric learning has been used to classify MOA by using a triplet loss"

I couldn't locate where the abbreviation *MOA* is initially introduced

- > "To date, the analysis of High-Content Screening assays is performed by handcrafted feature based analysis using classical image analysis software e.g. Cellprofiler "

Could you also include a reference to an existing work using Cellprofiler to perform HCS. That would make this statement complete in my opinion.

- > "with $k$ representing the number of archetypes, $n$ the number of data samples and $p$ the conditionality, respectively"

Could the symbols be elaborated more? What does *conditionality* imply - this wasn't introduced before in the text.

- > "Given input image I, we extract a feature map F by using a pretrained convolutional neural network c. Specifically, we take the last
flattened layer as feature map F ∈ R1×p where p denotes the number of feature dimensions."

What task was the convolutional network pretrained for? Which pretrained network was employed? Is the $p$ here the same as the conditionality introduced in Section 3. More details would be welcome.

- > "Here, it is particularly important that the rows of A and B are row-stochastic"

Could be followed with a general intuitive statement such as the entries of a row should sum to one.

- > "The triples consist of there different images: an anchor image f_a, a similar (positive) image f_p and a completely dissimilar (negative) image f_n

Small typo. Should be *triplets* instead of *triples*

- > "where α denotes the margin between positive and negative pairs. "

What is the magnitude of $\alpha$ chosen?

- In *Fig 2 (Right)*, the y-axis and x-axis labels can be renamed to $\text{tsne}_y$ and $\text{tsne}_x$

- Details about how baseline methods were executed would be welcome (For example, batch size, learning rate scheduling chosen in *SimCLR*, *EfficientNetB2* etc)

- "we can extract relevant and meaningful biological phenotypes using archetypal analysis, learn embeddings for visual inspection and for perfoming a pracftical top-down analysis"

Typo in *performing* and *practical*

**Final Rating After The Rebuttal:**

5: Strong Accept

**Justification Of The Final Rating:**

I thank the authors for addressing the suggestions. The updated manuscript and appendix reads much improved. I believe using archetype analysis to generate pseudo-labels is a novel approach and has significant, potential bio-medical application.

**Paper Type:**

methodological development

**Questions To Address In The Rebuttal:**

- Computer Vision Methods such as *SimCLR* often report performance of their method in terms of Top-1 and Top-5 Accuracy. While, using $\text{max} Z'$ is useful, could you include additional columns in Table 1 indicating the accuracy of baseline methods as well, as this is also an intuitive metric to parse?

- A general intuition on why methods such as *SimCLR*  performs so poorly would be welcome. For example, could you provide t-sne plots after training the *SimCLR* model?



**Special Issue:**

no

---

### Official Review · Reviewer_pRsm · 2022-01-24

**Confidence:** 2
**Preliminary Rating:** 4
**Recommendation:** Poster

**Summary:**

 The authors propose using a pretrained convolutional neural network to generate features for images and use archetype analysis to assign pseudo labels to these embeddings. These pseudo labels are then used to train an embedding network which uses a triplet loss based on the pseudo labels in order to generate a rich embedding network.

**Strengths:**

 The paper is well written and the explanation of archetype analysis is thorough and easy to understand. Presenting the alternative clustering based pseudo labeling helps motivate the use of archetype analysis

**Weaknesses:**

 The qualitative evaluation of the embeddings would be stronger if they were compared to the other benchmarked methods. Base lining against the embeddings from Figure 1.1 would help justify the use of using archetype analysis based pseudo labels or pseudo label generally.

**Deanonymize Review:**

no

**Detailed Comments:**

 Section 2 `and extension` -> `an extension, `formulations` -> `formulation`– typo. MOA abbreviation used but not defined. `transferred to knowledge of` potential typo?

The idea of using archetype analysis to create pseudo labels is interesting but the authors do not discuss what it means for the embedding network to learn from these pseudo labels. Would a perfect embedding network be able to separate images based on these pseudo labels? If so is it the case that the embedding network is essentially trying to force images into tighter clusters based on the clustered defined by the first feature extraction network in Figure 1.1? How much does the embedding richness depend on this first feature extractor?

What are the consequences of a “no fit” result in Table 1?


**Final Rating After The Rebuttal:**

4: Weak Accept

**Justification Of The Final Rating:**

I would like to thank the authors for updating the manuscript and addressing the comments in my review. However my concerns regarding the dependence on the initial embedding space have not been sufficiently addressed.

**Paper Type:**

methodological development

**Questions To Address In The Rebuttal:**

A more in-depth analysis into what the different embedding spaces represent and what the networks learn based on how the pseudo labels are created would be thought-provoking. Expanding on the discussion about why archetype analysis produces better pseudo labels than other methods would also be appreciated

**Special Issue:**

no

---

### Meta-Review · Area_Chair_9t4d · 2022-02-14

**Recommendation:** Accept (Poster)
**Confidence:** 5

**Metareview:**

This paper proposed a novel self-supervised learning method which uses archetypical analysis for image-based screening of cell culture models. There is agreement among the reviewers about the novelty in the use of archetypical analysis. The reviewers raised some initial questions in their comments, most of which were addressed by the authors in their rebuttal. Even though reviewers now agree that the paper should be accepted for MIDL, there are still points that need to be clarified in the camera ready, particularly the comment about the dependence on the initial embedding space raised by pRsm and the final comments listed by HiST. Overall, based on the reviewers recommendation, I'm happy to accept this work.

---

### Decision · Program_Chairs · 2022-02-28

Accept